# Interfacial Dynamics in Dual Channels: Inspired by Cuttlebone

**DOI:** 10.3390/biomimetics8060466

**Published:** 2023-10-01

**Authors:** Matthew Huang, Karl Frohlich, Ehsan Esmaili, Ting Yang, Ling Li, Sunghwan Jung

**Affiliations:** 1Department of Biological and Environmental Engineering, Cornell University, Ithaca, NY 14853, USA; mkh66@cornell.edu (M.H.); knf37@cornell.edu (K.F.); ee287@cornell.edu (E.E.); 2Department of Mechanical Engineering, Virginia Tech, Blacksburg, VA 24061, USAlingl@vt.edu (L.L.)

**Keywords:** interfacial dynamics, cuttlebone, Saffman-Taylor instability

## Abstract

The cuttlebone, a chambered gas-filled structure found in cuttlefish, serves a crucial role in buoyancy control for the animal. This study investigates the motion of liquid-gas interfaces within cuttlebone-inspired artificial channels. The cuttlebone’s unique microstructure, characterized by chambers divided by vertical pillars, exhibits interesting fluid dynamics at small scales while pumping water in and out. Various channels were fabricated with distinct geometries, mimicking cuttlebone features, and subjected to different pressure drops. The behavior of the liquid-gas interface was explored, revealing that channels with pronounced waviness facilitated more non-uniform air-water interfaces. Here, Lyapunov exponents were employed to characterize interface separation, and they indicated more differential motions with increased pressure drops. Channels with greater waviness and amplitude exhibited higher Lyapunov exponents, while straighter channels exhibited slower separation. This is potentially aligned with cuttlefish’s natural adaptation to efficient water transport near the membrane, where more straight channels are observed in real cuttlebone.

## 1. Introduction

The cuttlebone is a hard, brittle internal structure found in all members of the family Sepiidae, commonly known as cuttlefish. It is a chambered, water- and gas-filled shell used for buoyancy control [1,2,3,4]. The microscopic structure of the cuttlebone consists of narrow layers connected by numerous upright pillars [5,6,7,8]. The cuttlebone of cuttlefish aids in maintaining buoyancy by using its chambered structure to keep a gas mixture at a relatively constant pressure. This allows cuttlefish to adjust their buoyancy by regulating the gas-to-liquid ratio within the shell. Depending on the species, cuttlebones implode at a depth of 200 to 600 m [9,10].

A cuttlefish uses its cuttlebone as a rigid tank to control the buoyancy force. By pushing liquid in and out, the cuttlebone density can be easily adjusted, and buoyancy can be controlled during swimming [3]. The membrane attached to the cuttlebone can push the flow in and out of the cuttlebone chamber, controlling the osmotic pressure. This mechanism is different from what has evolved in fish, where changing the volume of the swim bladder is a key mechanism for buoyancy force. In Figure 1, we have shown the complex structure of the cuttlebone. The extensive nature with which the structure and composition of the cuttlebone have been studied has contributed to its frequent use as a model and inspiration.

As has been explained, the cuttlefish pumps liquid in and out of its cuttlebone structure to adjust its density and buoyancy force. Hence, the instability of the liquid-gas interface inside the cuttlebone can be a crucial factor in pumping. Any instabilities can result in forming droplets and bubbles inside the structure, and this can block the flow and reduce the efficiency of this process. In this study, we will investigate the different arrangements of slits with various degrees of waviness and lengths to understand the motion of the liquid-gas interface inside the cuttlebone chamber.

The manipulation of a liquid-gas interface within a narrow gap is a critical aspect of the buoyancy control mechanism in cuttlefish, involving the pushing and pulling of fluids in and out of their chambers. Researchers have explored the Saffman-Taylor instability to gain insights into the destabilization of this interface [11]. This phenomenon occurs when a less viscous fluid displaces a more viscous fluid within a confined channel, shedding light on the cuttlefish’s density adjustment mechanism. This instability is similar to what happens when water is drawn out of the narrow gap within the cuttlebone. Several mechanisms have been proposed to suppress the Saffman-Taylor instability, much like the challenges faced in industries where water is used to displace more viscous substances (e.g., petroleum) for extraction.

Researchers have been studying the effects of permeability gradients in a Hele-Shaw cell, a device used in fluid dynamics to study the behavior of fluids between two closely spaced parallel plates where there is a spatial gap width gradient [12,13,14], which also provides insights into cuttlebone’s buoyancy control. Experiments have been conducted to investigate suppression through tapering. It has been discovered that a negative taper can suppress the branching morphology. In addition, a more complex variant of the Hele-Shaw problem has been studied involving a thin, elastic membrane replacing the top plate [15,16,17], anisotropy of porous media [18], and more.

Furthermore, the role of solid vertical walls in affecting interface movement is significant. Several studies have examined how bubbles interact with rigid obstacles within a Hele-Shaw cell [19,20,21,22]. In microfluidic channels, the introduction of geometric obstacles has been employed to induce and regulate droplet breakups [23,24,25]. These phenomena can be theoretically explained using a one-dimensional Darcy flow model [26,27]. This model, which considers pressure drop along microfluidic channels, has practical applications in predicting and controlling the instability of air-liquid interfaces in microfluidic channels, offering insights relevant to phenomena such as the buoyancy control mechanism in cuttlefish.

In this paper, we will discuss the motion of a liquid-gas interface inside a channel inspired by the cuttlebone structure. Experimentally, different millimetric channels will be fabricated and tested to study the possible instability of the interface inside the cuttlebone. Studying liquid-gas interfaces within cuttlebones provides valuable insight into how fluid mechanics works at small scales, and it can be used as inspiration for further research projects involving similar phenomena across many fields.

## 2. Materials and Methods

### 2.1. Cuttlebone Samples

The cuttlebone has a chambered structure that consists of horizontal septa and vertical pillars. The horizontal septa divide the cuttlebone into separate chambers, and these chambers are supported by vertical pillars, which have a corrugated (or “wavy”) structure. The thickness of these pillars varies from species to species, but they are typically a few microns thick. The pillars extend vertically and are corrugated plates with compartments. Pillars are one of the main factors affecting the mechanical properties of the cuttlebone. The horizontal septa are typically thicker than the vertical pillars and consist of a double-layered structure. Overall, this chambered microstructure results in the cuttlebone having a porosity of over 90% by volume.

As depicted in Figure 1d–g, noticeable variations exist in the pillar shapes within the cuttlebone structure. Moving towards the dorsal side of the cuttlebone, the pillars exhibit a maze-like random orientation. However, near the membrane, in the siphuncular zone (the ventral side), the pillars are arranged parallel to one another and perpendicular to the membrane. This oriented structure is believed to play a facilitating role in water transport, ensuring effective filling of the cuttlebone.

### 2.2. Tests with Cuttlebone

In our study, we conducted two different tests to qualitatively observe water penetration within the cuttlebone. For the first, we took a cuttlebone and carefully cut it into a small cube. Placing the cuttlebone cube on the surface of blue-colored water, we observed the process of water penetration inside the cuttlebone. This experiment allowed us to visually assess how water interacts with and permeates the cuttlebone structure.

In the second test, we obtained several cuttlebone samples and dyed the ventral side of each sample. Subsequently, we submerged these dyed cuttlebone samples in water, immersing them to a depth of approximately 80 cm. Over varying periods of time, we carefully retrieved the submerged cuttlebone samples and performed longitudinal cuts down the center. By doing so, we could examine the extent of water penetration at different time intervals.

### 2.3. Channel Fabrication Using a 3D Printer

To facilitate a comprehensive analysis of this phenomenon, we have designed and conducted a series of experiments involving the fabrication of various fluidic channels with distinct geometries. These experiments allow us to gain insights into how different parameters, such as channel geometry and pressure, influence the behavior of liquid-gas interfaces.

In order to explore the effects of different channel arrangements, we employed a 3D printer (FormLabs 3) to create negative molds of the channels. The negative mold was printed using appropriate specifications. Subsequently, we mixed polydimethylsiloxane (PDMS) (Sylgard 184) with a curing agent, maintaining a ratio of 10 to 1. To ensure the removal of air bubbles, the PDMS mixture was placed in a vacuum chamber for an hour. The PDMS mixture was then poured onto the negative mold and cured for 5 h at 55 ℃.

To create the upper part of the microchannel, we coated a glass plate with a thin layer of PDMS using a spin coater, employing a rotation speed of approximately 1000 rpm for 1 min. Finally, the cured PDMS and glass plate were carefully attached and subjected to an additional 3 h of curing at 55 ℃. This fabrication process ensured the successful production of the desired channel configurations for subsequent analysis.

Each PDMS-glass plate consisted of two channels that ran approximately 3.9 cm long. The two channels were separated by a PDMS divider and split into three equally-sized segments. The separations between the segments reflected the non-continuous vertical walls of the cuttlebone, where small gaps between the vertical pillars aid in pressure equalization. The two channels of the PDMS-glass plate ran to a 1.4 cm by 0.9 cm rectangular head that was attached to our pressure apparatus. The head contained 12 circular PDMS pillars arranged in a 4 by 3 arrangement used to support it from collapsing inwards. This head served to equalize pressure at its interface with both channels.

### 2.4. Pushing the Air-Water Interface

To induce uniform movement of the liquid-gas interface within the channels, we employed a needle and a fluid bath accompanied by a vertical linear stage. By lowering the stage holding the fluid bath, we could achieve a wide range of pressure drops. The height difference between the channel and the fluid bath determines the pressure drop, following the equation ∆P = ρ g ∆H, where ∆P represents the pressure drop, ρ is the fluid density, g is the acceleration due to gravity, and ∆H signifies the height difference. Typically, the change in height could be achieved within a timeframe of less than 3 s, and the step was performed prior to the initiation of interface movement within the channel. Figure 2a–c provides a schematic representation of the experimental setup and the fabricated PDMS-glass plate, respectively.

In the design of our microfluidic channels, we consider a balance between inertia and viscous force. The Reynolds number represents a ratio of fluid inertia to viscous force, which is defined as ρUL/μ, where ρ is the fluid density (~997 kg/m^3^), U is the speed of the interface, L is the characteristic length scale, and μ is the dynamic viscosity of fluid (~1 × 10^−3^ Pa s). According to previous research [3], we estimate the speed of water movement within cuttlebone channels is approximately 300 μm/s within channels of about 100 μm. These parameters lead to an estimated Reynolds number of around 3 × 10^−2^ for the cuttlebone. In our designed artificial channels, we choose the median speed of the interface from 0.2 to 2 mm/s through 0.75-millimeter channels. Therefore, the corresponding Reynolds number range of 1.5 × 10^−2^ to 1.5 × 10^−1^ effectively encompasses the Reynolds number observed in the cuttlebone.

## 3. Results

### 3.1. Water Penetration in Cuttlebone

We have performed some experimental tests with real cuttlebones to examine their water penetration behaviors. Figure 3a illustrates the process of water penetration inside a small cuttlebone piece prepared by cutting it from a complete structure. This piece has both the top and bottom sides open without any sealing. By placing this cuttlebone piece in blue-dyed water, we observe a rapid water-rising phenomenon called capillary rise. This is primarily attributed to capillarity, which facilitates the movement of liquid through porous materials such as cuttlebone [28]. The momentum equilibrium governing the liquid flow within a porous substance is described by [29,30]:$$\rho \frac{d(z\;dz/dt)}{dt}=\frac{2\;\sigma \;cos\theta }{w}-\frac{\mu z}{w^{2}}\frac{dz}{dt}$$
where *z* is the column displacement, ρ is the water density, σ is the surface tension, *w* is the characteristic gap width, and μ is the water dynamic viscosity. This equation captures the interplay between inertia, surface tension, and viscous resistance. Initially, inertia is in equilibrium with surface tension, resulting in a prediction that the column height will undergo linear growth over time. As the flow gradually decelerates, the influence of viscous drag becomes more significant. Subsequently, the column height increases following a t^1/2^ relationship commonly known as the Washburn relation. In Figure 3b, it is evident that the majority of measurements align between t and t^1/2^. However, it is important to note that this capillary action is not the primary driving force behind fluid motion within a real cuttlebone. The cuttlebone’s fluid movement is primarily facilitated by osmotic pressure through the ventral membrane, highlighting a distinctly different mechanism.

Figure 4 illustrates the water penetration process into a submerged cuttlebone within a water tank. Our observations clearly indicated that water penetration into the cuttlebone increased proportionally with submersion duration. However, notable differences were observed when comparing the water penetration rate of a complete cuttlebone to that of the cuttlebone pieces shown in Figure 3, which were cut open. The slower rate of water penetration in the complete cuttlebone can be attributed to the presence of sealed air chambers within the structure. As water enters the cuttlebone, the air within these chambers becomes compressed, exerting resistance against further water penetration. Moreover, the water penetration process is non-uniform across different chambers, likely due to the inherent randomness of chamber distribution within the cuttlebone. These findings offer valuable insights into the water absorption capacity of the cuttlebone over time, providing qualitative information about the dynamic interaction between water and the intricate cuttlebone structure.

### 3.2. Motion of the Air-Water Interface in Artificial Channels

In this section, we will show the results of air-water interface movement within two artificial channels, drawing inspiration from the design of the cuttlebone. We varied the amplitude and pressure drops to examine how the waviness of the channel walls affects the movement of the air-water interface.

In Figure 5a,d, we present results from two linear channels, while Figure 5b,e shows two channels with an amplitude of 0.0625 mm, and Figure 5c,f represents two channels with an amplitude of 0.25 mm. All experiments were performed at the same pressure drop of 1.33 cm H_2_O. Notably, channels with more pronounced waviness exhibit faster motion of the front ends. Comparing the time taken for each channel to reach the end, the 0.25 mm amplitude channels required only about 21 s, while the linear channel took approximately 140 s. This suggests that wavy channels facilitate the suction of air into the channels. With the linear channels, the front ends initially start to separate but later converge with a small difference. However, as we increase the channel amplitude, we observe frequent crossings of the front ends. It is important to note that various random factors contribute to the movement of the channels.

Figure 6d–f illustrates the separation of two ends of channels corresponding to different pressure drops: 0.57 cm, 1.14 cm, and 1.33 cm. In this scenario, the front ends of channels consistently exhibit continuous separation over time without crossings. The separation time decreases as the pressure drop increases. This rate of separation motivates the use of the Lyapunov exponent method for characterization. To calculate the Lyapunov exponent, we measure the separation distance between the two front ends as D(t) = |S1(t) − S2(t)|, using the absolute value to eliminate bias towards a specific channel. The Lyapunov exponent is defined as the exponent of separation, λ, expressed as D(t) = exp(λt). We determine the Lyapunov exponent by calculating log(D(t))/t. However, the choice of time, t, is arbitrary in this analysis. Thus, we opt to compute the local slope of log(D(t)) over 7 data points using a fourth-order differentiation scheme implemented through a Matlab function called the ‘movingslope’, which applies a sliding window (e.g., a Savitzky-Golay filter) for differentiation.

In a single experiment, we obtained approximately 150 to 300 Lyapunov exponents over time. We then extract the upper values by considering only the 75th percentile of Lyapunov exponents, as shown in Figure 7. Positive values are crucial in Lyapunov exponent analysis as they indicate faster separation of the two front ends. Figure 7 presents the results of the 75th percentile of Lyapunov exponents. Two general trends can be observed. First, as the pressure drop increases, the Lyapunov exponents also increase. Higher pressure drops lead to faster motion of the air-water interface and, consequently, larger separations between the front ends. Second, the slope of the Lyapunov exponent against pressure drop is higher for curved channels and lower for straight channels. Additionally, in Figure 7b, channels with higher amplitudes exhibit higher Lyapunov exponents on average.

To understand fluid mechanics, it is important to consider three distinct pressure components: the capillary pressure drop, the viscous pressure drop, and the contact-line pressure drop.

The capillary pressure drop is attributed to the curvature of the frontal interface and is defined as 2γ/R, where γ is surface tension and R stands for the radius of curvature at the interface. In the context of dual channels, the radii of curvature remain identical. Consequently, there is no difference in capillary pressure drop between the two channels.

The viscous pressure drop, on the other hand, depends on the length of the channel filled with water, approximately μLU/h^2^, where μ is dynamic viscosity, L is the length of the liquid-filled channel, U is the speed of the interface, and h is the channel width/height. This pressure typically falls within the range of 10^−2^ to 10^−1^ Pa, which is smaller than the driving pressure (ρg∆H ~ 10–10^2^ Pa).

The third pressure drop originates from the contact line and is proportionate to μU/h. This pressure drop results from viscous dissipation due to the shearing of fluid between the wall and the meniscus around the contact point. This contact-line pressure drop is smaller than the viscous pressure drops, generally around 10^−3^ to 10^−2^ Pa.

Considering the above estimations, it becomes apparent that all three pressure components do not significantly contribute to driving the interface differently. The pressures at the two ends of the channels are nearly equal. On the air side, atmospheric pressure, denoted as P_atm_, is assumed. Conversely, at the head of the channels, the pressure is approximated as the pressure of the liquid bath, represented as P_atm_ − ρg∆H − 2γ/R, where ρ signifies fluid density, g represents the acceleration due to gravity, ∆H denotes the difference in height, and both the viscous pressure drop and the contact-line pressure drop are considered negligible. This lower pressure compared to the other side of the channel propels the interface towards the channel’s head.

The interfacial speed depends on the pressure difference across the channel length, as predicted by Darcy’s law. As the channel length decreases, the velocity further escalates, consequently amplifying the discrepancy in channel displacements. However, due to inherent irregularities in channels such as surface roughness and minor errors in channel shape, this difference may either intensify or attenuate. Hence, we need to use a statistical approach to characterize the dynamics.

## 4. Conclusions and Discussion

In this paper, we investigate the motion of liquid-gas interfaces within the cuttlebone to better understand fluid mechanics at small scales. We conduct experiments using various fluidic channels with distinct geometries to study the behavior of the liquid-gas interface inside the cuttlebone-like structure. The cuttlebone exhibits a chambered structure with horizontal septa and vertical pillars, where the microstructure plays a significant role in determining its mechanical properties. The fabricated channels are subjected to different pressure drops, and their motion is observed. Channels with more pronounced waviness demonstrate faster motion of the front ends, indicating that wavy channels enhance air suction. Additionally, we use the Lyapunov exponent method to characterize the separation of the front ends. The results reveal that higher pressure drops lead to increased Lyapunov exponents, indicating faster motion of the air-water interface and larger separations between front ends. Moreover, curved channels exhibit higher Lyapunov exponents compared to straight channels, and channels with higher amplitudes generally display higher Lyapunov exponents.

Our experiment finds a relationship between straighter channel structures and lower Lyapunov exponents (slower separation between the front ends of the air-water interface). Being that channel structures in the cuttlebone become more parallel with one another as they get closer to the membrane, our results suggest that this variation is to facilitate better water transport out of the cuttlebone. Meanwhile, for the majority of the cuttlebone structure, particularly in the region closest to the dorsal side, the walls are arranged in a labyrinth-like pattern. As shown previously, this more isotropic structure results in enhanced mechanical isotropy and structural performance. We conclude that cuttlefish and their cuttlebone channel structures may have developed structural adaptations locally to achieve desired functional requirements, including minimizing the instability for fluid transport as well as maximizing mechanical strength in selected regions.

This type of experimentation allows scientists to better understand the physical processes involved with liquids, which could potentially lead to advancements in both industry and medicine.

## Figures and Tables

**Figure 1 biomimetics-08-00466-f001:**
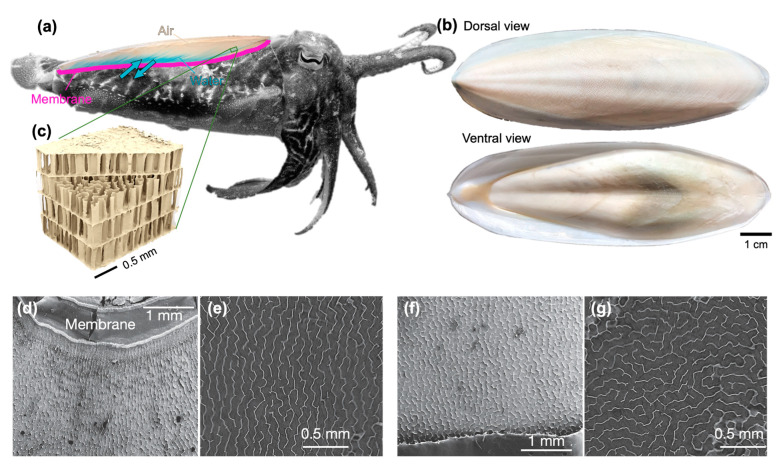
(**a**) Schematic of the cuttlebone inside the cuttlefish. Dyed blue water was used to visualize where and how water enters the cuttlebone. (**b**) Dorsal and ventral views of the cuttlebone. (**c**) The inset from Yang et al. [5] demonstrates the three-dimensional structures of the cuttlebone, illustrating its chambers and pillars. In the inset, the chambers run horizontally while the pillars run vertically. The top-viewed pillars of (**d**,**e**) a chamber near the membrane (the ventral side) and (**f**,**g**) a chamber closer to the dorsal side (**e**,**g**) illustrate similar regions as (**d**,**f**), respectively, with increased zoom.

**Figure 2 biomimetics-08-00466-f002:**
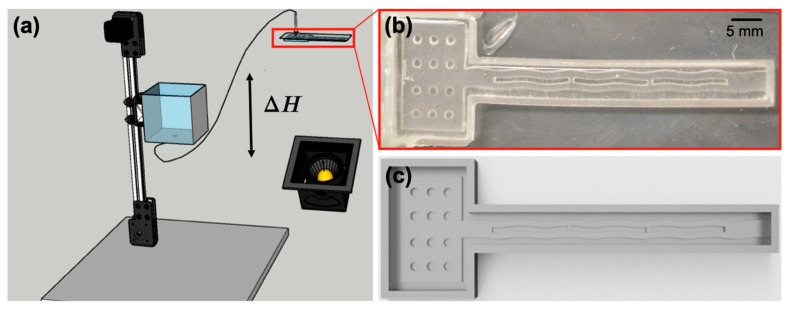
(**a**) Schematic of experiments; (**b**) zoomed-in photo; and (**c**) CAD drawing of a PDMS-glass plate.

**Figure 3 biomimetics-08-00466-f003:**
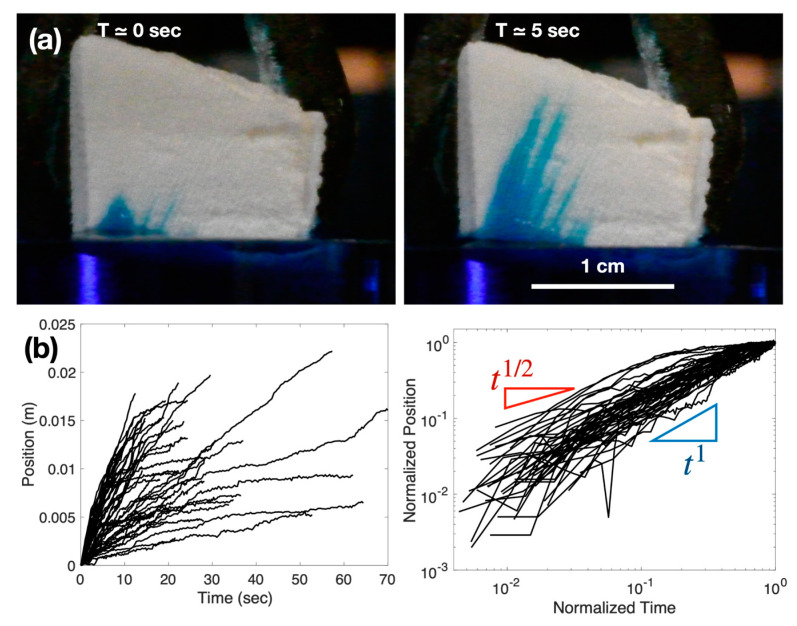
(**a**) Water penetration inside a cuttlebone. A small piece of cuttlebone with open top and bottom surfaces is in contact with blue-dyed water. Over time, the water gradually permeates the cuttlebone structure, providing insights into the absorption behavior within the material. (**b**) Water penetration length along the channel direction vs. time. The right panel is the plot with normalized time and displacement divided by its final time and displacement, respectively.

**Figure 4 biomimetics-08-00466-f004:**
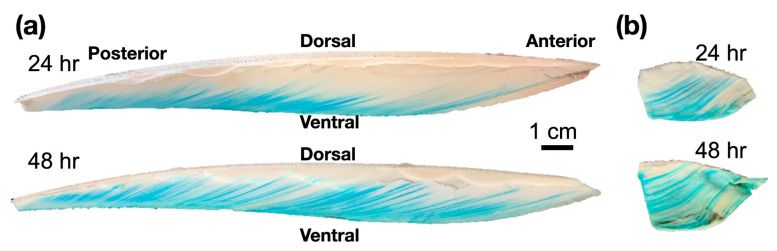
Water penetration in a complete cuttlebone. (**a**) Upper panel: after 24 h, and lower panel: after 48 h. The complete cuttlebone is sliced along the anterior-posterior direction in the middle, enabling observation of water penetration over time. (**b**) Upper panel: after 24 h, and lower panel: after 48 h. The cuttlebone is sliced along the left-right side direction, revealing the water penetration in a cross-sectional plane.

**Figure 5 biomimetics-08-00466-f005:**
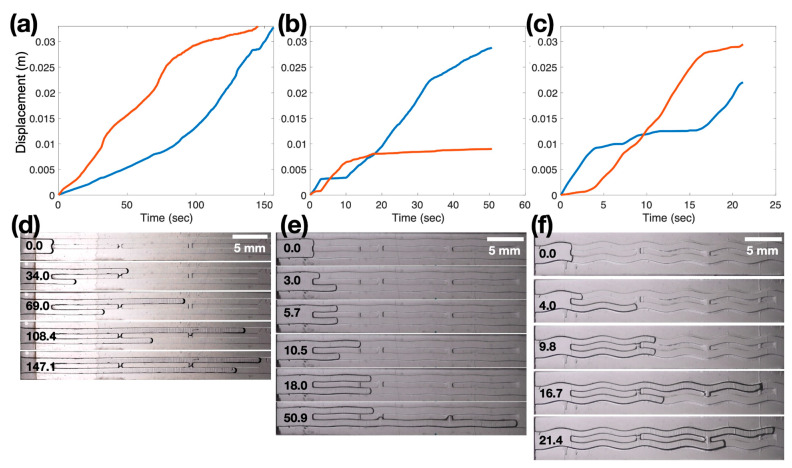
Displacements of two air-water interfaces in (**a**,**d**) straight channels or wavy channels of (**b**,**e**) 0.0625 mm amplitude or (**c**,**f**) 0.25 mm amplitude with a 1.33 cm H_2_O pressure drop. Panels (**a**–**c**) represent the displacements of the interfaces plotted against time, while panels (**d**–**f**) show corresponding image sequences of the channels. The red line and the blue line of panels (**a**–**c**) are the horizontal displacements of the top channel and the bottom channel, respectively.

**Figure 6 biomimetics-08-00466-f006:**
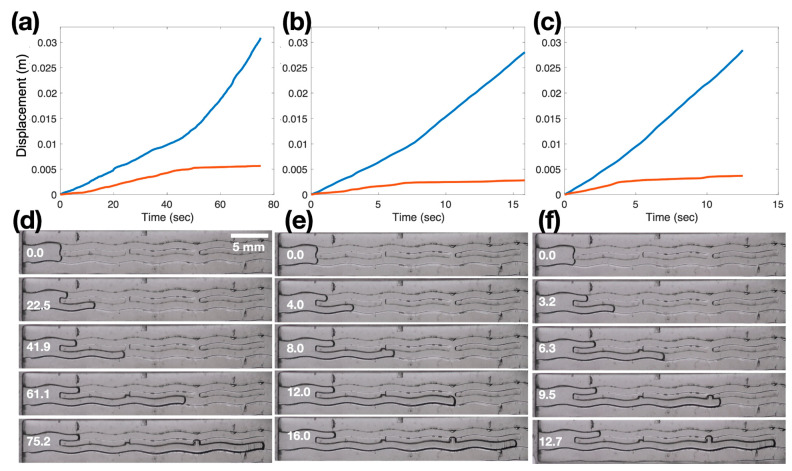
Displacements of two air-water interfaces in wavy channels of 0.188 mm amplitude with (**a**,**d**) 0.57 cm, (**b**,**e**) 1.14 cm, or (**c**,**f**) 1.33 cm pressure drop. Panels (**a**–**c**) represent the displacements of the interfaces plotted against time, while panels (**d**–**f**) show corresponding image sequences of the channels. The red line and the blue line of panels (**a**–**c**) are the horizontal displacements of the top channel and the bottom channel, respectively.

**Figure 7 biomimetics-08-00466-f007:**
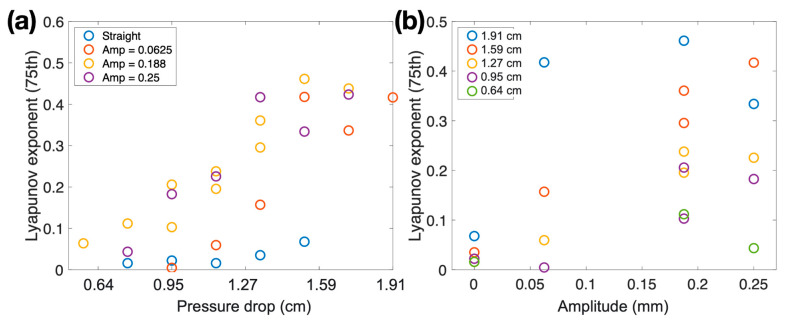
Lyapunov exponents of (**a**) different pressure drops and (**b**) different amplitudes.

## Data Availability

The data presented in this study are openly available at https://osf.io/kmb9y/ (accessed on 10-08-2023) at DOI 10.17605/OSF.IO/KMB9Y.

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
