# Peer review of "Interfacial Dynamics in Dual Channels: Inspired by Cuttlebone"

_biomimetics, 2023, doi:10.3390/biomimetics8060466_

Round 1

Reviewer 1 Report

This paper discusses the motion of a liquid-gas interface inside a channel inspired by a cuttlebone structure. Even though there are certain novelty, the paper needs further revise and improvement.

The comments are listed as follows:

(1)The background of this study is not clearly elaborated. What is the relationship between the interfacial dynamic characteristic of cuttlefish and Saffman-Taylor instability? the explanation of each literature is too simple.  Section of introduction should provide a detailed and comprehensive explanation.

(2) This study lacks scientific rigor. It is advisable to begin by identifying several factors that influences the interfacial dynamic characteristic and then employ methods such as DOE or RSM for a systematic investigation. This would help ascertain the influences of various crucial parameters.

(3) There are some grammar errors, it is recommended to refinish language carefully.

(4) How can artificial channels accurately simulate the structure of cuttlefish, more details should be provided.

(5)For Fig5 and Fig 6, what do the red line and blue line mean?The content of the figures needs to be thoroughly explained.

There are some grammar errors, it is recommended to refinish language carefully.

Author Response

Reviewer #1

This paper discusses the motion of a liquid-gas interface inside a channel inspired by a cuttlebone structure. Even though there are certain novelty, the paper needs further revise and improvement.

The comments are listed as follows:

(1)The background of this study is not clearly elaborated. What is the relationship between the interfacial dynamic characteristic of cuttlefish and Saffman-Taylor instability? the explanation of each literature is too simple.  Section of “introduction” should provide a detailed and comprehensive explanation.

Thank the reviewer for the comment. Cuttlefish manipulate their buoyancy by pushing liquid in and out of the cuttlebone structure. As the less viscous air displaces water when water is pulled out, instability would ensue without the proper evolutionary adaptations.

To better explain the relationship between Saffman-Taylor instability and the cuttlebone structure, we added the text, “Likewise, when water is pumped out from the cuttlebone, the less viscous air displaces more viscous water, and this is when we might expect the Saffman-Taylor instability to occur inside the chamber.” 

(2) This study lacks scientific rigor. It is advisable to begin by identifying several factors that influences the interfacial dynamic characteristic and then employ methods such as DOE or RSM for a systematic investigation. This would help ascertain the influences of various crucial parameters.

Thank the reviewer for the comment. To match fluid-dynamic characteristics of artificial channels to the cuttlebone, we check the Reynolds number, which represents the ratio of inertia to viscous stress. The cuttlebone has the Reynolds number is about 3X10-2, while our experiments with artificial channels are in a range of Re = 1.5X10-2 to 1.5X10-1. In the revised manuscript, we included one paragraph like 

“In the design of our microfluidic channels, we consider a balance between the inertia and viscous force. The Reynolds number represents a ratio of fluid inertia to viscous force, which is defined as ρUL/μ where ρ is the fluid density (~997 kg/m3), U is the speed of the interface, L is the characteristic lengthscale, and μ is the dynamic viscosity of fluid (~1X10-3 Pa s).  According to previous research [3], we estimate the speed of water movement within cuttlebone channels is approximately 300 μm/s within channels of about 100 μm. These parameters lead to an estimated Reynolds number of around 3X10-2 for the cuttlebone. In our designed artificial channels, we choose the median speed of the interface from 0.2 to 2 mm/s through 0.75-mm channels. Therefore, the corresponding Reynolds number range of 1.5X10-2 to 1.5X10-1, effectively encompasses the Reynolds number observed in the cuttlebone.”  

(3) There are some grammar errors, it is recommended to refinish language carefully.

We went through the manuscript and corrected any grammatical errors.

(4) How can artificial channels accurately simulate the structure of cuttlefish, more details should be provided.

Thank the reviewer for the comment. 

In addition to matching the Reynolds number between the cuttlebone and artificial channels, we added detailed descriptions of the channel geometry. In short, our channel is designed to simplify the complex structure in the cuttlebone. The wavyness is varied by controlling the amplitude of the channels, and the speed is controlled by changing the height of the water bath. Also, vertical walls in the cuttlebone are not continuous, which is reflected in the artificial channels by having small separations in between. 

Our newly added paragraph is “COPY AND PASTE”

(5)For Fig5 and Fig 6, what do the red line and blue line mean?The content of the figures needs to be thoroughly explained.

Thank the reviewer for the comment. 

To clarify this, we added text like “The red line and the blue line of panels (a, b, c) are the horizontal displacements of the top channel and the bottom channel, respectively. This description has been added to both figures.” 

Reviewer 2 Report

This work is mainly experimental work. The authors described the experimental setup and results very clearly. However, there was a lack of theoretical analysis to explain the experimental observations. Without further discussion of the experimental data, the application of the results found this manuscript to other channel geometries and flows is very limited. 

Author Response

Reviewer #2

Thank the reviewer for the comment. Following the suggestion, we conducted a comprehensive analysis to estimate the pressure drop contributions from capillary pressure, viscous stress, and contact-line dissipation. Our findings indicate that none of these factors can account for the observed discrepancy between the two interfaces within the channels.

Nonetheless, in accordance with Darcy's law, it becomes evident that as the channel length decreases, the velocity proportionally increases, thereby magnifying the disparities in channel displacements.

In the revised manuscript, we added several paragraphs as below. 

To understand the fluid mechanics, it is important to consider three distinct pressure components: the capillary pressure drop, the viscous pressure drop, and the contact-line pressure drop.

The capillary pressure drop is attributed to the curvature of the frontal interface and is defined as 2γ/R, where γ is surface tension, and R stands for the radius of curvature at the interface. In the context of dual channels, the radii of curvature remain identical. Consequently, there exists no difference in capillary pressure drop between the two channels.

The viscous pressure drop, on the other hand, depends on the length of the channel filled with water, approximately μLU/h2, where μ is dynamic viscosity, L is the length of the liquid-filled channel, U is the speed of the interface, and h is the channel width/height. This pressure typically falls within the range of 10-2 to 10-1 Pa, which is smaller than the driving pressure (ρg∆H ~ 10 - 102 Pa). 

The third pressure drop originates from the contact line and is proportionate to μU/h. This pressure drop results from viscous dissipation due to the shearing of fluid between the wall and the meniscus around the contact point. This contact-line pressure drop is smaller than the viscous pressure drop, generally around 10-3 to 10-2 Pa.

Considering the above estimations, it becomes apparent that all three pressure components do not significantly contribute to driving the interface differently. The pressures at the two ends of the channels are nearly equal. On the air side, atmospheric pressure, denoted as Patm, is assumed. Conversely, at the head of the channels, the pressure is approximated as the pressure of the liquid bath, represented as Patm - ρg∆H - 2γ/R, where ρ signifies fluid density, g represents the acceleration due to gravity, ∆H denotes the difference in height, and both the viscous pressure drop and the contact-line pressure drop are considered negligible. This lower pressure compared to the other side of the channels propels the interface towards the channel's head.

The interfacial speed depends on the pressure difference across the channel length, as predicted by Darcy’s law. As the channel length decreases, the velocity further escalates, consequently amplifying the discrepancy in channel displacements. However, due to inherent irregularities in channels such as surface roughness and minor errors in channel shape, this difference may either intensify or attenuate. Hence, we need to use the statistical approach to characterize the dynamics. “

Round 2

Reviewer 1 Report

Sorry but I still think the reply is too simple. the background of this study is not clearly elaborated. More literature should be added and analyzed in this paper.

For the previous question 2,  the authors only indtroduced one parameter:Reynolds number. More infuencing factors should be included.

Author Response

Reviewer #1

Sorry but I still think the reply is too simple. the background of this study is not clearly elaborated. More literature should be added and analyzed in this paper.

Thank the reviewer for the comment. In response, we have substantially expanded the Introduction to provide a more comprehensive overview and context. 

For the previous question 2,  the authors only indtroduced one parameter:Reynolds number. More infuencing factors should be included.

Thank the reviewer for the comment. We agree that there is indeed a need to clarify the relative significance of various factors contributing to instability. To address this, we have added several additional paragraphs explaining the minor role played by surface tension and other viscous dissipation effects in the observed phenomena. Consequently, we have emphasized that other non-dimensional numbers such as the Weber number, Bond number, and Ohnesorge number are not crucial to the understanding of this specific instability. Below is the refined paragraph added to the 'Results' section for your consideration: 

To understand the fluid mechanics, it is important to consider three distinct pressure components: the capillary pressure drop, the viscous pressure drop, and the contact-line pressure drop.

The capillary pressure drop is attributed to the curvature of the frontal interface and is defined as 2γ/R, where γ is surface tension, and R stands for the radius of curvature at the interface. In the context of dual channels, the radii of curvature remain identical. Consequently, there exists no difference in capillary pressure drop between the two channels.

The viscous pressure drop, on the other hand, depends on the length of the channel filled with water, approximately μLU/h2, where μ is dynamic viscosity, L is the length of the liquid-filled channel, U is the speed of the interface, and h is the channel width/height. This pressure typically falls within the range of 10-2 to 10-1 Pa, which is smaller than the driving pressure (ρg∆H ~ 10 - 102 Pa). 

The third pressure drop originates from the contact line and is proportionate to μU/h. This pressure drop results from viscous dissipation due to the shearing of fluid between the wall and the meniscus around the contact point. This contact-line pressure drop is smaller than the viscous pressure drop, generally around 10-3 to 10-2 Pa.

Considering the above estimations, it becomes apparent that all three pressure components do not significantly contribute to driving the interface differently. The pressures at the two ends of the channels are nearly equal. On the air side, atmospheric pressure, denoted as Patm, is assumed. Conversely, at the head of the channels, the pressure is approximated as the pressure of the liquid bath, represented as Patm - ρg∆H - 2γ/R, where ρ signifies fluid density, g represents the acceleration due to gravity, ∆H denotes the difference in height, and both the viscous pressure drop and the contact-line pressure drop are considered negligible. This lower pressure compared to the other side of the channels propels the interface towards the channel's head.

The interfacial speed depends on the pressure difference across the channel length, as predicted by Darcy’s law. As the channel length decreases, the velocity further escalates, consequently amplifying the discrepancy in channel displacements. However, due to inherent irregularities in channels such as surface roughness and minor errors in channel shape, this difference may either intensify or attenuate. Hence, we need to use the statistical approach to characterize the dynamics. “

Reviewer 2 Report

The revision is now acceptable. 

Author Response

Thank you so much for providing constructive feedback.